# Patient complexity assessment tools containing inquiry domains important for Indigenous patient care: A scoping review

**Anika Sehgal** [1]*, **Cheryl Barnabe**[2], **Lynden (Lindsay) Crowshoe**[3]

**1** Department of Community Health Sciences, Cumming School of Medicine, University of Calgary, Calgary, Alberta, Canada, **2** Department of Medicine, Cumming School of Medicine, University of Calgary, Calgary, Alberta, Canada, **3** Department of Family Medicine, Cumming School of Medicine, University of Calgary, Calgary, Alberta, Canada

* anika.sehgal@ucalgary.ca

## Abstract

Patient complexity assessment tools (PCATs) are utilized to collect vital information to effectively deliver care to patients with complexity. Indigenous patients are viewed in the clinical setting as having complex health needs, but there is no existing PCAT developed for use with Indigenous patients, although general population PCATs may contain relevant content. Our objective was to identify PCATs that include the inquiry of domains relevant in the care of Indigenous patients with complexity. A scoping review was performed on articles published between 2016 and 2021 to extend a previous scoping review of PCATs. Data extraction from existing frameworks focused on domains of social realities relevant to the care of Indigenous patients. The search resulted in 1078 articles, 82 underwent full-text review, and 9 new tools were identified. Combined with previously known and identified PCATs, only 6 items from 5 tools tangentially addressed the domains of social realities relevant to Indigenous patients. This scoping review identifies a major gap in the utility and capacity of PCATs to address the realities of Indigenous patients. Future research should focus on developing tools to address the needs of Indigenous patients and improve health outcomes.

## Introduction

Generally agreed to be a separate entity from comorbidity or multi-morbidity [1], patient complexity [2–4] is deemed to arise from the social and contextual factors that impact health outcomes [1,5,6]. The complex interplay between the various determinants of health and their presentation within patients can be difficult to address for healthcare providers (HCPs) [2,7]. Patient complexity assessment tools (PCATs) are screening tools that have been proposed as a means to aid HCPs in collecting vital information to identify the source of complexity and to effectively deliver care to patients [8–10].

While there are a number of PCATs that have been developed to meet the needs of various patient populations, there is no such tool to address the health needs of Indigenous patients. Stemming from the longstanding impacts of colonization, structural inequities within health,

cihr-irsc.gc.ca/e/193.html) grant number: FDN
143284 awarded to CB. The funders had no role in
study design, data collection and analysis, decision
to publish, or preparation of the manuscript.

**Competing interests:** The authors have declared
that no competing interests exist.

education, and social service systems continue to limit the capacity of Indigenous peoples to
pursue good health [11–13]. HCPs today rarely comprehend the full scope of the historical
and ongoing social drivers of poor health that impact Indigenous patients arising from multi-
generational impacts of colonization, but rather demonstrate an overall lack of awareness and
competency that translates into ineffective care [14–16]. To date, there is no existing PCAT
developed for use with Indigenous patients, although general population PCATs may contain
relevant content.

Developed to improve HCP communication and clinical approaches when providing care
to Indigenous persons with diabetes [11], the 'Educating for Equity' or 'E4E' framework out-
lines Indigenous-specific determinants of care. The E4E framework includes a comprehensive
assessment of the social realities that contextualize an individual's capacity to cope with their
condition and has since been applied to arthritis care providers' continuing medical education
[17]. These realities include social and economic resource disparities, and the accumulation of
adverse life experiences [11]. Social and economic resource disparity is a normalized state for
many Indigenous peoples—with limited choices and stress, one's capacity to pursue healthy
behaviors is affected [11]. This can further be aggravated within family contexts that result in
the diversion of resources among many people [11]. Furthermore, health knowledge can be
limited through structural barriers and conflicts that arise from relationships with HCPs, fur-
ther contributing to disparities impacting one's state of health [11]. The accumulation of
adverse life experiences includes assessing family and community adversity arising from his-
torical trauma and poverty, multiple forms of loss (personal and collective) due to coloniza-
tion, and the intergenerational impact of residential schools [11,18]. In the E4E Framework,
facilitators are also recognized. "Culture framing knowledge" refers to knowledge contextuali-
zation and exchange in a manner that is effective in building a shared understanding with the
patient [11,17]. Finally, recognizing "culture as therapeutic" is acknowledging that health is
positively correlated with a secure cultural identity while having access to cultural resources,
including traditional healing practices and medicine for Indigenous peoples [11].

To date, there is no existing PCAT developed that incorporates the Indigenous-specific
determinants of health including social and economic resource disparities and adverse life
experiences as outlined by the E4E care framework. Having PCATs that address complexity
among Indigenous patients arising from longstanding and permeating impacts of colonization
[19,20] and identify appropriate pathways to health equity [21] is in alignment with the Truth
and Reconciliation Commission of Canada's Calls to Action [22]. Furthermore, the extent to
which existing PCATs address and engage the realities of Indigenous patients remains
unknown since general population PCATs may still contain relevant content. Therefore, we
undertook this scoping review to investigate existing PCATs and how inclusive they are of the
social realities for Indigenous patients, based on an existing framework, 'Educating for Equity'
[11].

## Methods

### Study design and search strategy

A scoping review was purposefully chosen to identify existing PCATs due to the emergent
nature of patient complexity and its evolving conceptualization [7,23] and was developed and
is reported according to the Preferred Reporting Items for Systematic Reviews and Meta-Anal-
yses extension for Scoping Reviews [24]. The protocol was developed a priori replicating the
search strategy from a previous review by Marcoux et al. [25] which was determined to be reli-
able and thorough [26] to curate available screening tools to identify patients with complex
health needs needing frequent care. As those authors conducted their search in 2016 we

restricted our search between January 1st 2016 and April 8th 2021 to identify any newly developed and published PCATs using two databases, CINAHL and Scopus (which is inclusive of EMBASE and MEDLINE). Reference lists of identified articles were also scanned for additional relevant studies. Title/abstract screening and full-text review was conducted independently by two reviewers (AS, EB) and conflicts were resolved by discussion and consensus.

### Inclusion and exclusion criteria

Studies were included if they were in the English language, presented a questionnaire or screening tool to identify patient complexity for an adult population, consistent with the inclusion criteria identified by Marcoux et al. [25]. Studies were excluded if they were not in English, limited to specific populations of psychiatric, pediatric, and pregnant women, were designed for a specific disease/illness, or were focused on predictive modelling based on insurance claims, aligned with Marcoux et al. [25] again. In addition to this exclusion criteria, studies that used a compilation of several tools to assess complexity or that were modifications of an existing tool but adapted for a specific disease/illness were not retained in our search. Studies were extracted to Covidence software for screening. Authors who discussed a questionnaire or screening tool but did not include it in the full-text of their article were contacted and requested to provide the tool if possible.

### Data extraction and analysis

A pre-tested data extraction form was used to record a selected article's identifying information, population, intervention, and evaluation outcomes. Each item or question from included PCAT's (those identified by Marcoux et al. [25] and in our search) were extracted and compiled into an item pool. Each item or question was then categorized by which domain they were related to—these domains were derived from a review of common domain across the tools by the authors, informed by domain categorization as defined in the development of one of the most commonly employed PCATs [27], and further confirmed by previous research that identified parallel themes [28]. Domains included assessments of biological/physical states, social/socioeconomic status, psychological state, access to healthcare services, patient health literacy, and the patient's ability to function independently. It was also noted if the tool addressed any aspects of the social realities, including barriers and facilitators, as described and categorized in the E4E care framework for a total of four domains [11]. These included social and economic resource disparities, the accumulation of adverse life experiences, culture framing knowledge, and culture being therapeutic [11]. Items were assigned to domains based on qualitative analysis techniques known as "descriptive" coding whereby codes are characterized by conceptual unity [29]. Each item was only assigned to one domain which it represented most closely. All items were reviewed by an expert group to ensure reliability and accuracy of assignment.

### Results

The search strategy returned a total of 1668 articles, 1294 from Scopus and 374 from CINAHL. After removing 590 duplicates, a total of 1078 articles were included, 82 were identified to potentially be included following title/abstract screening, and 9 were included after a full-text review by two reviewers (see Fig 1). A total of 3 authors were contacted to provide their instrument, including those who were identified in the previous review by Marcoux et al. [25], the overall response rate was 67%. By combining previously known and identified PCATs with new PCATs developed after the scoping review by Marcoux et al. [25], a total of 18 tools were analyzed in this scoping review.

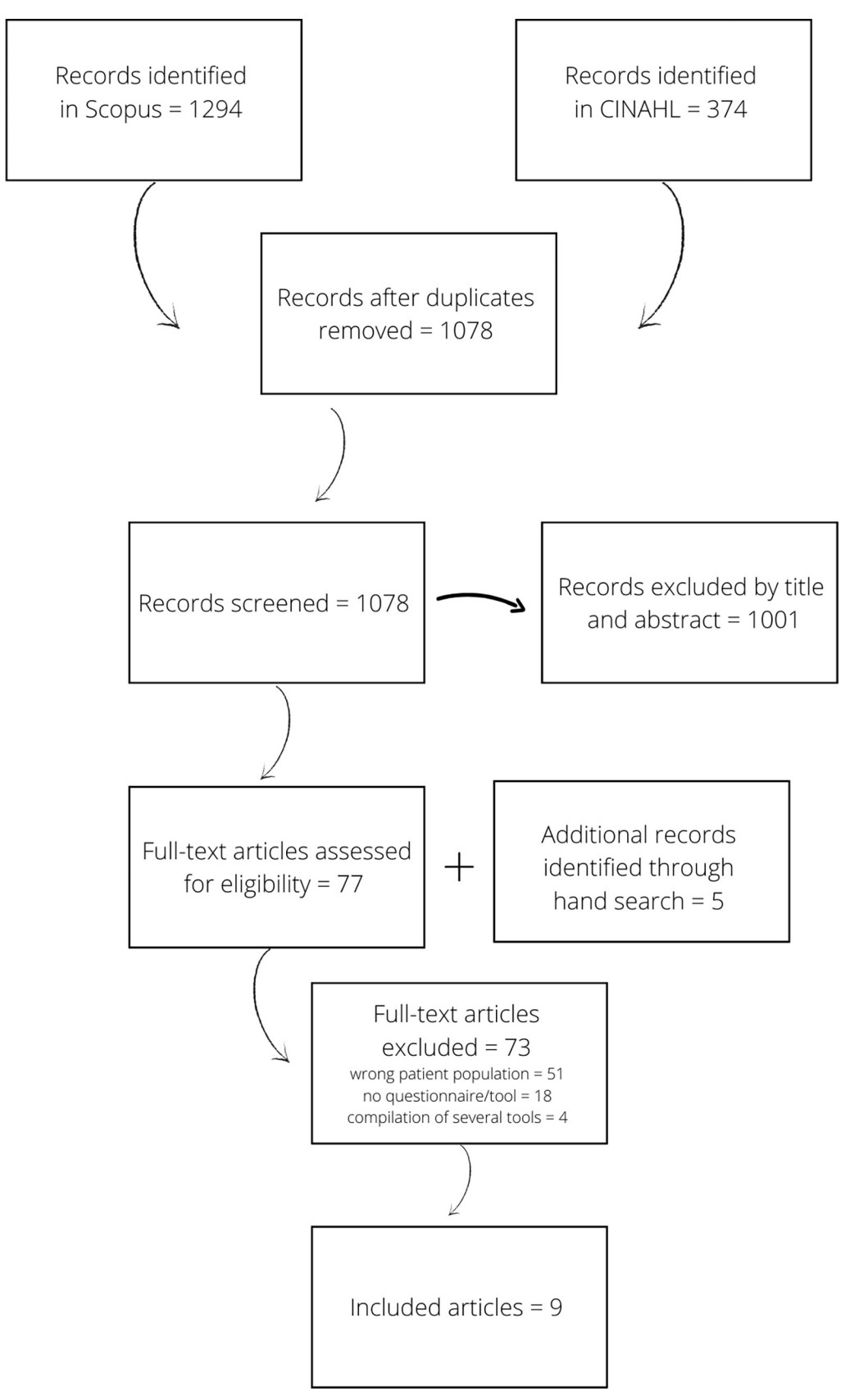

**Fig 1. Scoping review flow diagram.**

### New study characteristics

A total of 9 new instruments were identified by extending the search strategy of Marcoux et al. [25], these included the MCAM [2]; the PCAM [10]; CONECT-6 [30]; homelessness and underutilization health service questions [31]; supporting the support system questions [32]; the questions used in a case management collaborative community program [33], the COM-PRI [34]; the MECAM [35]; and the OCCAM [36]. Table 1 displays the study characteristics of the new instruments identified by this scoping review including their target population, mode of administration, and intended outcome.

### Characteristics and domains of identified tools

Table 2 displays the characteristics of the 18 included tools, including the domains of inquiry [2,10,27,30–44]. Studies were published between 1998 to 2021, and included at least 3 items with the maximum number of items being 42. A total of 13 instruments were administered by a HCP [2,10,27,32–37,40,41,43,44], 3 tools were self-administered [30,31,42], and 2 tools were completed in collaboration between the HCP and patient [38,39]. While all included tools were employed in an adult population, 6 instruments specified their use for the elderly population [39–44].

### Items that engaged social realities

From the 18 tools included in this review, over 300 items were extracted. There were 67 items assigned to the healthcare access domain, 57 items assigned to the biological/physical domain, 49 items assigned to the social/socioeconomic status domain, 23 items assigned to the psychological domain, 17 items assigned to the health literacy domain, and 9 items assigned to the functionality domain. A number of items were deemed to fit into more than one domain: for example, 12 items assessed both biological/physical and functionality domains, 8 items assessed both the healthcare access and functionality domains, and 6 items addressed both the

**Table 1. Novel instruments identified by the scoping review.**

| Name of instrument or short title | Population characteristics | Mode of administration for the instrument | Intended outcome |
|---|---|---|---|
| COmplex NEeds Case-finding Tool-6 (CONECT-6) [30] | Adult patients with chronic conditions | Self-report | Identify adult patients with ambulatory care sensitive conditions and complex health needs in emergency departments |
| Homelessness and underutilization health service questions [31] | Homeless people | Self-report | Utilization of different types of health services in the past six months. |
| Supporting the support system [32] | Adult patients and their caregivers | Healthcare provider/case manager | Assess needs of the patients' support system |
| A Collaborative Community Program in Remote Northern Territory [33] | Aboriginal people in rural and remote areas of the Northern Territory of Australia | Healthcare provider/case manager | Identify issues which contribute to patient visiting ED and identify solutions |
| MCAM [2] | Adult patients | Healthcare provider | Identify the factors that may be interfering with the care of a patient |
| Patient Centered Assessment Method (PCAM) [10] | Adult patients | Healthcare provider | Identify any biopsychosocial complexities that are impacting the patient |
| COMPRI [34] | Adult patients | Healthcare provider | Identify and facilitate interdisciplinary care coordination for patients |
| MECAM [35] | Adult patients | Healthcare provider | Identify any factors that are posing a risk to the well-being of a patient |
| OCCAM [36] | Adult patients | Healthcare provider | Facilitate care coordination for a patient with complex health needs |

**Table 2. Domains assessed by identified tools and questionnaires.**

| Name of instrument or short title | Purpose | # of items[a] | Domains assessed[b] | | | | | | |
|---|---|---|---|---|---|---|---|---|---|
| | | | Bio. | Social | Psych. | HC access | Health literacy | Function. | Social realities |
| A Collaborative Community Program in Remote Northern Territory [33] | Identify issues which contribute to patient visiting ED and identify solutions | 21+ | ✓ | ✓ | ✓ | ✓ | ✓ | ✓ | ✓ |
| Homelessness and underutilization [31] | Assess health service usage among homeless people | 5+ | | | | ✓ | | | |
| Supporting the support system [32] | Assess patients and their support systems | 7 | | ✓ | ✓ | ✓ | | ✓ | |
| CONECT-6 [30] | Identify patients with complex needs | 6 | ✓ | ✓ | ✓ | ✓ | | ✓ | |
| Homeless Screening Risk of Re-Presentation [37] | Identify homeless people at risk of re-hospitalization | 8 | ✓ | ✓ | ✓ | ✓ | | ✓ | |
| Pie [38] | Identify workers at high risk of healthcare expenditure | 10 | ✓ | | ✓ | ✓ | | | |
| Reuben [39] | Identify high risk of hospitalization among adults | 10 | ✓ | ✓ | | ✓ | | ✓ | ✓ |
| ARORA [40] | Identify adults at risk of hospitalizations | 42 | ✓ | ✓ | | ✓ | ✓ | ✓ | |
| Initial Assessment Interview Questions [41] | Identify high risk seniors | 35 | ✓ | ✓ | ✓ | ✓ | ✓ | ✓ | |
| Pra [42] | Identify risk of hospital admission | 8 | ✓ | ✓ | | ✓ | | ✓ | |
| SIGNET TRST [43] | Improve case finding and coordinate care | 6 | ✓ | ✓ | ✓ | ✓ | | ✓ | |
| INTERMED [27] | Indicate need for multidisciplinary care | 20 | ✓ | ✓ | ✓ | ✓ | | ✓ | |
| MCAM [2] | Identify factors interfering with care | 10 | ✓ | ✓ | ✓ | ✓ | | ✓ | ✓ |
| MECAM [35] | Identify factors posing risk to patient well-being | 11 | ✓ | ✓ | ✓ | ✓ | ✓ | ✓ | |
| OCCAM [36] | Facilitate care coordination | 27 | ✓ | ✓ | ✓ | ✓ | | ✓ | ✓ |
| CARS [44] | Identify elders at risk of hospitalization | 3 | ✓ | | | ✓ | | | |
| PCAM [10] | Identify biopsychosocial complexities | 12 | ✓ | ✓ | ✓ | ✓ | ✓ | ✓ | |
| COMPRI [34] | Indicate need for interdisciplinary care coordination | 10 | ✓ | ✓ | ✓ | ✓ | | ✓ | |

[a]The number of items with a + indicate the minimum items asked at baseline with the potential of additional items depending on responses gathered.

[b]Domains investigated are the biological, social, psychological, healthcare access, health literacy, functionality, and social realities.

biological/physical and social/socioeconomic status domains. Over 15 items fit into more than 2 domains, with the maximum being 5 domains. Items that were aimed at collecting administrative data were not assigned to any domains. Only 6 items spanning across 5 different tools were assessed to engage the social realities of Indigenous patients (see Table 3 and Fig 2). From the OCCAM [36], two items tangentially addressed the social realities of Indigenous patients: these were (i) "HCPs are to assess adverse influence of others within the last two weeks regarding patient's health related behaviour" and (ii) "HCPs are to assess childhood past history including disrupted parenting, abuse, and disrupted schooling." Both of these items indirectly address aspects of adverse life experiences that shape health. From the MCAM [2], one item partially addressed how culture frames knowledge to build a shared understanding of health, "HCP to assess patient's shared language and culture with provider." From Reuben [39], the patient's participation in religious services is being assessed, this item partially addresses how culture is therapeutic and correlated with good health. From the INTERMED [27], the HCP is assessing whether or not the patient has any resistance to treatment, this item is partially addressing if and how any past adverse life experiences have contributed to their resistance. Finally, from the case management collaborative community program [33], a cluster of items

**Table 3. Items engaging social realities.**

| Name of instrument or short title | Item from instrument | Engagement with social realities |
|---|---|---|
| OCCAM [36] | HCP to assess adverse influence of others within the last two weeks regarding patient's health related behaviour | This item partially addresses aspects of adverse life experiences that shape health |
| | HCP to assess childhood past history including disrupted parenting, abuse, and disrupted schooling | This item partially addresses aspects of adverse life experiences that shape health |
| MCAM [2] | HCP to assess patient's shared language and culture with provider | This item partially addresses how culture frames knowledge to build a shared understanding of health |
| INTERMED [27] | HCP to assess if the patient has any resistance to treatment | This item partially addresses aspects of adverse life experiences that may have contributed to the resistance to treatment |
| Reuben [39] | Patient to self-report if less than weekly participation at religious services | This item partially addresses how culture is therapeutic and correlated with good health |
| A Collaborative Community Program in Remote Northern Territory [33] | *Food security*:<br>Did you go hungry yesterday?<br>How many times did you go hungry in the last week?<br>Do you worry about how you will get your next meal?<br>Do you worry that people will steal your food?<br>Does anyone give you free meals? | These items partially address aspects of social and economic resource disparity that shape health |

addressed food security by asking "Did you go hungry yesterday? How many times did you go hungry in the last week? Do you worry about how you will get your next meal? Do you worry that people will steal your food? and Does anyone give you free meals?" Together, these items address food security which is just one aspect of social and economic resource disparities that shape health outcomes. Fig 2 displays the items identified that address the social realities of Indigenous patients.

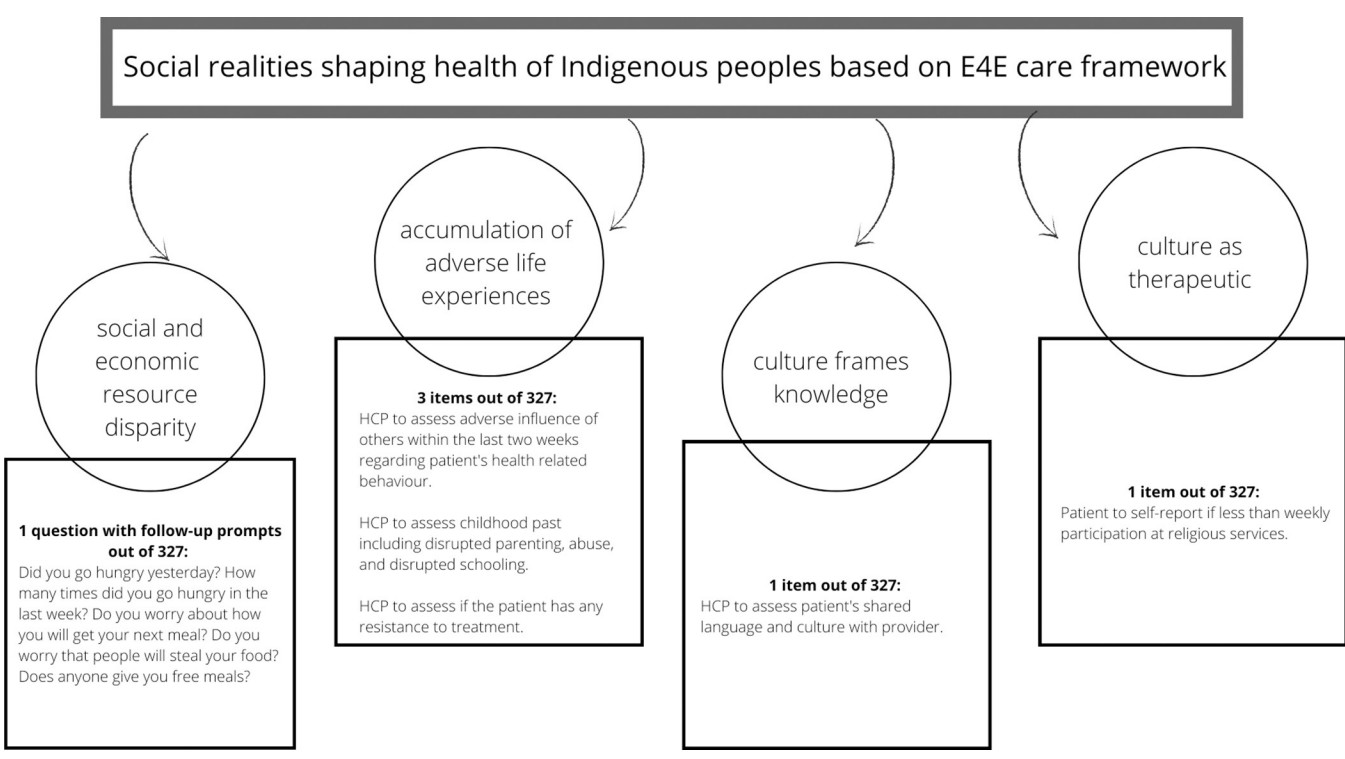

**Fig 2. Items addressing social realities.**

## Discussion

The purpose of this review was to identify and describe existing PCATs to determine the extent to which they are applicable to Indigenous patients and to establish the need for an Indigenous-centered PCAT. By replicating the search strategy outlined by Marcoux et al. [25], we identified an additional 9 PCATs yet none of these comprehensively addressed the factors Indigenous peoples, highlighting a continuous failure to address the social realities that are known to shape the health of Indigenous peoples, as demonstrated in the lack of items identified in Table 3 and Fig 2. Without an explicit investigation into the sources that cause Indigenous patients to present with complexity, existing PCATs unable to identify or address the ways in which to reduce Indigenous patient complexity. Recommendations to address social realities are presented below.

### Social and economic resource disparity

Exploring socioeconomic limitations and acknowledging the effect of resource disparities is a means to identify and understand why an Indigenous patient is presenting with complexity [11,45]. Items to assess social and economic resource disparities should be inclusive of an individual's direct resources but rather note how these disparities have manifested themselves at the familial and community levels, and in turn, impacted their health. PCATs should allow the HCP to locate sources of complexity that are not directly present within the patient but rather the broader domains shaped by colonization [46] that continuously undermine their health. Social disparities can also operate at a knowledge level through inequities that manifest themselves in educational systems and frameworks, furthering social trauma and directly impacting health literacy. Therefore, it is essential that PCATs that aim to assess complexity among Indigenous patients include items that explore health literacy from the perspective of Indigenous peoples.

### Adverse life experiences

If we are to understand the "complexity" of Indigenous patients, acknowledging and identifying how psychosocially mediated adverse life events, rooted in a legacy of colonialism, have shaped their health is essential [16,47]. Only one item from pre-existing PCATs assessed childhood history (see Table 3), yet for Indigenous peoples, the recent history of residential schools continues to undermine the health of their population. The cultural genocide perpetrated by residential schools and other colonial policies has significantly harmed the health and wellbeing of individuals, families, and entire communities [48–51]. If a PCAT is to identify the source of complexity among Indigenous patients, a respectful investigation into specific psychosocial adverse life experiences, including the impacts of historical trauma, is necessary.

### Culture frames knowledge

From an Indigenous cultural lens, knowledge is relational within healthcare interactions, therefore, all knowledge, including the patient's and the HCP's, should be respected, shared, and exchanged. Building deeper therapeutic relationships is achieved through this cultural lens, and a patient's access to and comprehension of health-related knowledge is facilitated [11,20,52]. Despite this, for many Indigenous patients, healthcare experiences are often plagued by racism and discrimination [53], drawing parallels to their historical experiences, further deterring them from seeking care [16]. PCATs that aim to assess complexity among Indigenous patients should include items that assess the extent to which the HCP is able to

connect with the patient at a relational level, demonstrating cultural humility and cultural safety [54,55].

### Culture as therapeutic

Connecting with cultural resources and *ways of doing* is considered to be therapeutic in nature —reconnecting with Indigenous identity and ceremony not only builds resilience but also promotes a sense of social cohesion [11,52]. In healthcare settings, accessing Indigenous healing modalities may be important to Indigenous patients yet they are continuously undermined due to the longstanding impacts of colonization. If a PCAT is to assess complexity among Indigenous patients, it should be culturally congruent in that it assesses health as it is defined in Indigenous worldviews and should include items that explore the patient's preferences in regards to traditional medicine and healing practices along with their ability to access and effectively utilize these resources.

### Pre-existing domains across PCATs

Though there were no such tools that comprehensively engaged the social realities relevant to Indigenous populations, pre-existing domains readily assessed in instruments to date should not be discredited completely. These domains should be contextualized to discern the layers of historical trauma and ongoing injustices that have largely shaped the health of Indigenous peoples. Recommendations to contextualize pre-existing domains in PCATs are presented in Table 4.

### Additional considerations for complexity among Indigenous patients

While recognizing and integrating social realities into PCATs for Indigenous patients is important to identify and address the root causes of complexity, there are cautions that must

**Table 4. Pre-existing domains contextualized for Indigenous patients.**

| Domain | What it includes | What it is missing for Indigenous patients |
|---|---|---|
| Biological/physical | • Assessment of biological and physical disease/concerns including co- or multi-morbidity | • Higher rates of diseases, lower life expectancies, and higher levels of infant and maternal mortality [56] along with increased likelihood of co- and/or multi-morbidities [57] and mental health manifestations of physical disease [58]. |
| Social/SES | • Assessment of socioeconomic status, social factors that shape health | • Social determinants of health shaped by legacy of colonialism including forced assimilation, displacement, and lifestyle changes [45].<br>• how these determinants, including housing, education, employment, food security, and access to clean drinking water can cause complexity [45]. |
| Psychological/ Emotional | • Assessment of mental health and emotional status | • Impacts of historical and ongoing trauma including intergenerational trauma, collective loss and grief, and structural violence that have manifested themselves into mental health problems including higher rates of suicide [59–61]. |
| Healthcare Access | • Assessment of physical ability to access healthcare services<br>• Assessment of healthcare coverage | • Separate and complex systems of healthcare for status versus non-status FNMI peoples, on-reserve versus off-reserve FNMI peoples [15,45,46,61] along with diffusion of responsibility among governmental entities to provide healthcare to Indigenous peoples.<br>• Leading to complications in access, safety, and quality that contributes to "complexity." |
| Health Literacy | • Assessment of individual's ability to comprehend health information, make appropriate decisions | • Historical events have created a mistrust in the healthcare system and ongoing racism, communication barriers, and stereotypes reinforce this mistrust, limiting capacity to obtain health literacy [62–64]. |
| Functionality | • Assessment of individual's independence and autonomy, degree to which support is required to function on a regular basis | • Disability caused by fear, mistrust, and avoidance of care, including social supports as they have oppressed, mistreated, and endangered Indigenous peoples [65–67].<br>• Disability in community and/or family roles such as not being able to fulfill cultural responsibilities [68]. |

be addressed to prevent PCATs from perpetuating power imbalances, re-traumatizing Indigenous patients, and addressing complexity from an exclusively Western worldview. Power dynamics have been theorized to be the driving force behind inequities in healthcare [69] whereby both overt and implicit discrimination and racism create barriers to healthcare access for Indigenous peoples [70–72]. PCATs developed for Indigenous patients should not reinforce stereotypes [71,73] or "other" Indigenous peoples [74]; items should be framed from a decolonial lens and be asked by the HCP from a place of humility [75,76] while embodying principles of cultural safety [77]. Furthermore, PCATs for Indigenous peoples should abstain from making Pan-Indigenous assumptions but rather give space to respect diversity within Indigenous groups. Given that complexity among Indigenous patients arises from the traumatic inter-generational and multi-level impacts of colonization and can manifest itself through several various different pathways [78], it is important that questions are framed in a manner that seeks permission before exploring social realities, for example asking *"is it okay if we talk about your living conditions?"* allowing the patient to control the depth of information they feel comfortable disclosing. PCATs to date are conceptualized from Western ideologies in that they locate the complexity at the individual level and approach complexity from a deficit-narrative [79]. PCATs for Indigenous patients should identify how social injustices rooted in a colonial legacy have shaped the health outcomes of Indigenous peoples while not hyper-focusing on where deficiencies exist but rather recognizing the resilience and strengths of Indigenous peoples [78,80]. By incorporating an assessment of resilience and protective factors that prevent complexity, a PCAT for Indigenous patients may provide an avenue for HCPs to recognize the rich legacy of Indigenous strengths, work alongside the patient to advance Indigenous health equity, and acknowledge the dominance of Western health models [81].

## Strengths and limitations

To the best of our knowledge, this is the first scoping review to explore PCATs within the context of application to Indigenous populations while specifically investigating the domains of existing PCATs and how they map onto the social realities that continue to shape the health outcomes of Indigenous patients. Furthermore, we have identified an additional 9 tools since the last scoping review of PCATs, presenting a recent list of pre-existing instruments that aim to assess complexity. One limitation of this study is that it did not evaluate the quality of articles being included, however, scoping reviews typically do not evaluate study quality which is aligned with standard recommendations [82]. Another limitation of this review is that it may have missed non-English studies due to inclusion/exclusion criteria, along with any unpublished studies residing in the grey literature. Furthermore, since this review replicated a previously developed search strategy, articles that were potentially relevant but omitted in the previous scoping review by Marcoux et al. [25] may have been missed again—our hand search did identify an additional 5 tools to be included, though none of these effectively engaged the social realities of Indigenous patients. It is worth noting that no PCAT included in this scoping review described in-depth collaboration or consultation with the patient population for whom it was being developed and/or employed. This lack of patient engagement prompts the need for future research to incorporate the voices of patients themselves so that the sources of complexity, including the social realities, are accurately described and measured in clinical tools such as PCATs. Respectful and appropriate patient engagement has been at the forefront of Indigenous health research for several years now [83,84] and as such, any attempt at developing a PCAT for Indigenous patients should include the voices and lived experiences of Indigenous patients themselves.

## Conclusion

The present review has not identified any pre-existing tools that have been developed to identify and assess the social realities that shape the health of Indigenous patients who present with complexity. While there are several PCATs that are inclusive of general population needs, the factors most relevant to Indigenous populations remain unattended to. Although there were select items from few tools that inadvertently tapped into some of the social realities, they are not comprehensive nor are they applicable as stand-alone items to appropriately identify and address patient complexity among Indigenous peoples. Furthermore, the clinical utility of these items, even if combined, remains unknown and unclear, with little potential to demonstrate validities that should be present in clinical tools. Overall, this review highlights that PCATs to date have neglected to include domains relevant to Indigenous patients, reflecting an insidious pattern imbedded within colonial systems of healthcare. Results of this scoping review will be used to ground and inform future work that aims to develop a PCAT for Indigenous patients.

## Supporting information

**S1 File. PRISMA-ScR checklist.**
(PDF)

## Acknowledgments

The authors would like to thank Elaine Boyling who played a critical role in title/abstract screening and full-text reviews.

## Author Contributions

**Conceptualization:** Anika Sehgal.

**Data curation:** Anika Sehgal.

**Formal analysis:** Anika Sehgal.

**Investigation:** Anika Sehgal.

**Methodology:** Anika Sehgal, Cheryl Barnabe, Lynden (Lindsay) Crowshoe.

**Supervision:** Cheryl Barnabe, Lynden (Lindsay) Crowshoe.

**Validation:** Anika Sehgal.

**Visualization:** Anika Sehgal.

**Writing – original draft:** Anika Sehgal.

**Writing – review & editing:** Anika Sehgal, Cheryl Barnabe, Lynden (Lindsay) Crowshoe.

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
