## [Decision Letter · Decision Letter 0]

14 Jun 2022

PONE-D-21-39743Patient complexity assessment tools containing inquiry domains important for Indigenous patient care: A scoping review.PLOS ONE

Dear Anika Sehgal,

Thank you for submitting your manuscript to PLOS ONE. After careful consideration, we feel that it has merit but does not fully meet PLOS ONE’s publication criteria as it currently stands. Therefore, we invite you to submit a revised version of the manuscript that addresses the points raised during the review process.

We look forward to receiving your revised manuscript.

Kind regards,

Jennifer Annette Campbell

Academic Editor

PLOS ONE

**Journal requirements:**

Reviewers' comments:

Reviewer's Responses to Questions

**Comments to the Author**

1. Is the manuscript technically sound, and do the data support the conclusions?

Reviewer #1: Yes

Reviewer #2: Yes

2. Has the statistical analysis been performed appropriately and rigorously? 

Reviewer #1: Yes

Reviewer #2: N/A

3. Have the authors made all data underlying the findings in their manuscript fully available?

Reviewer #1: Yes

Reviewer #2: Yes

4. Is the manuscript presented in an intelligible fashion and written in standard English?

Reviewer #1: Yes

Reviewer #2: Yes

5. Review Comments to the Author

Reviewer #1: ***Please see attachment to see information in clearer format

--Summary / General Commentary--

Concerning the content of the review, no major errors were apparent. Under close scrutiny, the inquiry was both factually accurate and offered a balanced, skilfully coherent interpretation of prior art. Areas for improvement are relatively inconsequential, rarely warranting more than a few additional lines of analysis or a graphic to deepen the interpretation of findings.

--Positive Remarks--

1. This review achieved an impressive balance between technicality and understandability, allowing it to remain informative and non-obvious to future readers with a wide range of expertise.

2. All data and discussions were communicated firmly. Explanations of the circumstances experienced by indigenous populations are well-developed and informed, never compromising on accuracy or tact.

3. Among many others, the “Additional Considerations” section (particularly the connections that it drew to deficit-centric narratives) was sharp, insightful, and diverse - demonstrating a commendable level of nuance and depth of exploration on the part of the authors.

Recommendations:

--Domains of Inquiry--

The introduction of your study suggests that all PCATs were primarily assessed on their agreement with the facets of E4E.

Furthermore, lines 75-76 state that the E4E framework included “social and economic resource disparities, and the accumulation of adverse life experiences,” Given that these factors are the basis for much of the analysis that will be conducted in the review, they should be covered in more depth and in their entirety from the outset.

Additionally, the process by which it is decided that an item in a given PCAT aligns with E4E or not is not immediately clear. Along with the mention of the expert group that evaluated the validity of each category, this can be supplemented with hypothetical examples or related devices:

“Patient to self-report if they are religious” - Example of tangential agreement with “culture as therapeutic” requirement because regular practice of religion is not captured.

vs.

“Patient to self-report less than weekly participation at religious services” - Example of full agreement with “culture as therapeutic” requirement because regular practice is captured.

Such examples would better illustrate why certain items in a PCAT would be judged as agreeing, disagreeing, or partially agreeing to the fundamental aspects of E4E, and would make the panel feel less like a black box.

--Highlighting the Influence of Recent PCATs--

As outlined in table 1, it appears that by reiterating Marcoux’s protocol in the present day, nine additional complexity assessment tools were identified.

These PCATs must be further emphasized, since the value proposition of this review hinges (to a large degree) on how these new instruments affect the original conclusion drawn by Marcoux et al.

More specifically, they provide important context into PCATs that simply did not exist when Marcoux published her research in 2017, and could realistically be used to delineate a continued failure to consider indigenous social realities in the past half-decade.

Whether the domains of inquiry in these new instruments seem to support Marcoux’s original conclusion or oppose it in some respects, the changes they warrant to the existing consensus should be more prominent in the “Discussion” or “Conclusion”.

--Methodology Rationale--

Little rationale was provided for why the current review followed the same methodology as that of Marcoux et al.

Although it is clear among indigenous health researchers who have read that Marcoux’s methodology was reliable and exhaustive, this is not necessarily a given for readers outside the field. All considered, this could be resolved quite directly with some additional commentary in the Protocol section that explains why the same protocol was continued.

--Synthesizing Bird’s-Eye Trends--

Might there be domains of inquiry that are consistently underrepresented across many of the PCATs that were analyzed? Although this is a scoping review, gaining a more numerical understanding of exactly which dimensions are lacking will lead to a more definitive response and render the conclusion more precise.

More emphasis should be placed on inter-test trends - both to make your inquiry more enriching, and to better meet your stated aim of assessing the want for a dedicated PCAT for indigenous populations.

In the case of your review, an extra graph that summarizes the cumulative exposure each E4E factor cumulatively receives from all the PCATs would contribute greatly to the tangibility of the findings, beyond the general claim that indigenous social realities are lacking.

--Making Conclusions More Comparative--

Stating that MCAM, OCCAM, and other PCATs largely neglect indigenous social realities is a good start. However, this leaves gaps for further questioning, since a small (but non-trivial) portion of the community still believes that PCATs perform poorly in general - whether the patient is indigenous or not.

To negate any such counterpoint, it would be useful to compare the alignment of each instrument with both indigenous and non-indigenous populations. Your review makes a more substantial claim: not only do current PCATs neglect domains relevant to indigenous patients; they do so while being relatively inclusive of non-indigenous patients. In this way, you can highlight a disparity.

But to do this, this part of your analysis must be two-sided. This does not call for any major reworking, save for a few more lines in either the discussion or conclusion to make it clear that there is evidence of one group being neglected in these assessments.

At the very least, it would emphasize that a considerable amount of forethought was directed into this inquiry.

--Miscellaneous notes--

In the OCCAM section of table 3 (pg. 15) it appears that the word “of” is missing in “This item partially addresses adverse life experiences that shape health”.

--Final Remarks --

It has been some time since I last reviewed a paper in this specific niche. This was a welcome and refreshing piece that I genuinely look forward to reading following publication, and whose potential for social good and reconciliation is especially compelling.

Reviewer #2: Thank you for the opportunity to review this article. The article is a scoping review identifying the potential utility of using items in existing patient complexity assessment tools (PCATs) within indigenous populations and need for an indigenous-centered PCAT and is an extension of a previous scoping review of PCATs. Nine new PCATs were identified through the review (18 total), and of these existing PCATs, only 6 of 300 total individual items were deemed partially relevant to indigenous populations. The manuscript is well-written and makes the case for need of an indigenous-centered PCAT. Comments to consider to further strengthen the manuscript are included below.

Introduction

• How do PCATs, and specifically an indigenous-specific (or other cultural groups/identities) PCAT, differ from culturally informed assessments or interviews, including the DSM-5’s Cultural Formulation Interview? What benefits do PCATs (and indigenous-specific PCAT) provide over something like the CFI?

• Consider moving information from the “framework for indigenous patient complexity domains” section into the introduction section; this seems more like background material than method material. Doing so would provide relevant background information about the social determinants of health that uniquely impact indigenous populations and contribute to complexity, setting the stage for why a PCAT is needed.

Method/Figures

• The numbers in figure 1 do not seem to add up. For example, after excluding 1001 records from the total 1078 records screened, it seems that 77 full text articles should have been assessed for eligibility (not 82), and of those, 9 records should have been identified (82 total minus 73 excluded; not 4). Please review for accuracy or clarify the presented numbers.

Results/Discussion

• It seems that there is already an existing PCAT for aboriginal Australians (i.e., A Collaborative Community Program in Remote Northern Territory), which hits on each of the 7 domains of interest. A discussion of how/why this 21-item PCAT is not sufficient as an indigenous-specific PCAT is needed.

• As stated by the authors in the discussion, PCATs seem to approach the individual from a deficit-narrative. How might an indigenous-specific PCAT take into consideration cultural factors that are protective and may be utilized to promote resilience?

• Acknowledging that each indigenous group has a unique culture and may have variations in structural issues contributing to complexity, how might an indigenous-centered PCAT generalize? Would it be applicable to all indigenous groups across the world, or would there need to be adaptations made to fit each group? This ties into the earlier comment about use of the already existing Collaborative Community Program in Remote Northern Territory.

Minor

• Please review for typos, missing or repeated words, etc.

---

## [Author Response · Author response to Decision Letter 0]

30 Jun 2022

Response to Reviewer Recommendations

Reviewer #1: 

Domains of Inquiry: The introduction of your study suggests that all PCATs were primarily assessed on their agreement with the facets of E4E. Furthermore, lines 75-76 state that the E4E framework included “social and economic resource disparities, and the accumulation of adverse life experiences,” Given that these factors are the basis for much of the analysis that will be conducted in the review, they should be covered in more depth and in their entirety from the outset. 

We agree there is better positioning by making this change. We have moved information from the “framework for Indigenous patient complexity domains” to the introduction section. Please see lines 61 to 84 which now explain the E4E framework from the outset of the manuscript. We have also changed the following paragraph to reflect the importance of the E4E care framework, please see lines 85-94. 

Additionally, the process by which it is decided that an item in a given PCAT aligns with E4E or not is not immediately clear. Along with the mention of the expert group that evaluated the validity of each category, this can be supplemented with hypothetical examples or related devices: “Patient to self-report if they are religious” - Example of tangential agreement with “culture as therapeutic” requirement because regular practice of religion is not captured. vs. “Patient to self-report less than weekly participation at religious services” - Example of full agreement with “culture as therapeutic” requirement because regular practice is captured. Such examples would better illustrate why certain items in a PCAT would be judged as agreeing, disagreeing, or partially agreeing to the fundamental aspects of E4E, and would make the panel feel less like a black box.

Unfortunately, we did not assess items based on the extent to which they address the E4E framework components. We only assessed items in a binary fashion, as to whether or not they addressed an aspect of the E4E framework. In the case of Indigenous patients, more appropriate questions would ask their connection to their culture, their relationship with spirituality, and the extent to which they feel connected to Indigenous worldviews, values, and beliefs. This manuscript is part of a larger research project which is developing a PCAT for Indigenous patients which is indicated in our conclusion, as such these considerations will be taken into account as we progress. 

Highlighting the Influence of Recent PCATs: As outlined in table 1, it appears that by reiterating Marcoux’s protocol in the present day, nine additional complexity assessment tools were identified. These PCATs must be further emphasized, since the value proposition of this review hinges (to a large degree) on how these new instruments affect the original conclusion drawn by Marcoux et al. More specifically, they provide important context into PCATs that simply did not exist when Marcoux published her research in 2017, and could realistically be used to delineate a continued failure to consider indigenous social realities in the past half-decade.

Whether the domains of inquiry in these new instruments seem to support Marcoux’s original conclusion or oppose it in some respects, the changes they warrant to the existing consensus should be more prominent in the “Discussion” or “Conclusion”.

We have further emphasized the continuous failure to consider Indigenous social realities in the first paragraph of our “discussion” section with the following sentence: “By replicating the search strategy outlined by Marcoux et al. [25], we identified an additional 9 PCATs yet none of these comprehensively addressed the factors Indigenous peoples, highlighting a continuous failure to address the social realities that are known to shape the health of Indigenous peoples, as demonstrated in the lack of items identified in Table 3 and Fig 2.” Please see lines 259 to 263. 

Methodology Rationale: Little rationale was provided for why the current review followed the same methodology as that of Marcoux et al. Although it is clear among indigenous health researchers who have read that Marcoux’s methodology was reliable and exhaustive, this is not necessarily a given for readers outside the field. All considered, this could be resolved quite directly with some additional commentary in the Protocol section that explains why the same protocol was continued.

Thank you for this suggestion. There was a need to update the search to confirm if additional PCATs were available for consideration, as described in our methods. We reflect on the reliability of the search strategy and have clarified this in the following sentence: “The protocol was developed a priori replicating the search strategy from a previous review by Marcoux et al. [25] which was determined to be reliable and thorough [26] to curate available screening tools to identify patients with complex health needs needing frequent care.” Please see lines 147-150. 

Synthesizing Bird’s-Eye Trends: Might there be domains of inquiry that are consistently underrepresented across many of the PCATs that were analyzed? Although this is a scoping review, gaining a more numerical understanding of exactly which dimensions are lacking will lead to a more definitive response and render the conclusion more precise. More emphasis should be placed on inter-test trends - both to make your inquiry more enriching, and to better meet your stated aim of assessing the want for a dedicated PCAT for indigenous populations. In the case of your review, an extra graph that summarizes the cumulative exposure each E4E factor cumulatively receives from all the PCATs would contribute greatly to the tangibility of the findings, beyond the general claim that indigenous social realities are lacking.

We attempted to demonstrate this in Figure 2 and Table 4, but with this comment we interpret that additional enhancements to the figure were indeed necessary. Please see the revised Figure 2. 

Making Conclusions More Comparative: Stating that MCAM, OCCAM, and other PCATs largely neglect indigenous social realities is a good start. However, this leaves gaps for further questioning, since a small (but non-trivial) portion of the community still believes that PCATs perform poorly in general - whether the patient is indigenous or not. To negate any such counterpoint, it would be useful to compare the alignment of each instrument with both indigenous and non-indigenous populations. Your review makes a more substantial claim: not only do current PCATs neglect domains relevant to indigenous patients; they do so while being relatively inclusive of non-indigenous patients. In this way, you can highlight a disparity. But to do this, this part of your analysis must be two-sided. This does not call for any major reworking, save for a few more lines in either the discussion or conclusion to make it clear that there is evidence of one group being neglected in these assessments. At the very least, it would emphasize that a considerable amount of forethought was directed into this inquiry.

We have added two sentences in our conclusion section to make our claims more comparative and substantial. These sentences are as follows: “While there are several PCATs that are inclusive of general population needs, the factors most relevant to Indigenous populations remain unattended to” and “Overall, this review highlights that PCATs to date have neglected to include domains relevant to Indigenous patients, reflecting an insidious pattern imbedded within colonial systems of healthcare.” Please see lines 372-373 and 378-380 respectively. 

Miscellaneous notes: In the OCCAM section of table 3 (pg. 15) it appears that the word “of” is missing in “This item partially addresses adverse life experiences that shape health”.

Thank you for bringing this to our attention, we have added the word “of” now. 

Reviewer #2: 

Introduction: How do PCATs, and specifically an indigenous-specific (or other cultural groups/identities) PCAT, differ from culturally informed assessments or interviews, including the DSM-5’s Cultural Formulation Interview? What benefits do PCATs (and indigenous-specific PCAT) provide over something like the CFI?

Thank you for this comment. PCATs are generally agreed to be screening tools or starting points to assess the source(s) of complexity. Culturally informed assessments and interviews may be tools employed after the initial screening of a patient who is presenting with complexity. We have highlighted this in our introduction with changes to the following sentence: Patient complexity assessment tools (PCATs) are screening tools that have been proposed as a means to aid HCPs in collecting vital information to identify the source of complexity and to effectively deliver care to patients. Please see lines 48 to 50. 

Consider moving information from the “framework for indigenous patient complexity domains” section into the introduction section; this seems more like background material than method material. Doing so would provide relevant background information about the social determinants of health that uniquely impact indigenous populations and contribute to complexity, setting the stage for why a PCAT is needed.

This was also suggested by the other reviewer, and we agree, thus have moved information from the “framework for Indigenous patient complexity domains” to the introduction section. Please see lines 61 to 84 which now explain the E4E framework from the outset of the manuscript. We have also changed the following paragraph to reflect the importance of the E4E care framework, please see lines 85-94. 

Method/Figures: The numbers in figure 1 do not seem to add up. For example, after excluding 1001 records from the total 1078 records screened, it seems that 77 full text articles should have been assessed for eligibility (not 82), and of those, 9 records should have been identified (82 total minus 73 excluded; not 4). Please review for accuracy or clarify the presented numbers.

Thank you for pointing out the discrepancy. In the box ‘Full text articles assessed for eligibility’ we had included both those retained from the title and abstract selection (n=77) and the additional records identified through references and hand searching (n=5). We have revised this now, please see Figure 1 and lines 190-193. 

Results/Discussion: It seems that there is already an existing PCAT for aboriginal Australians (i.e., A Collaborative Community Program in Remote Northern Territory), which hits on each of the 7 domains of interest. A discussion of how/why this 21-item PCAT is not sufficient as an indigenous-specific PCAT is needed. 

This tool was interesting to us as well, but it was focused on identifying issues which contribute to ED visits rather than all domains of complexity. Further, we assessed it was not inclusive of all social and economic resource disparities, nor is it inclusive of the other 3 domains of the E4E framework, therefore not comprehensive of all the realities faced by Indigenous peoples. 

As stated by the authors in the discussion, PCATs seem to approach the individual from a deficit-narrative. How might an indigenous-specific PCAT take into consideration cultural factors that are protective and may be utilized to promote resilience?

This is an excellent proposal, and will be forming our approach in the development of an Indigenous-specific PCAT. We have discussed this further under the “additional considerations for complexity among Indigenous patients” and added the following sentence: “By incorporating an assessment of resilience and protective factors that prevent complexity, a PCAT for Indigenous patients may provide an avenue for HCPs to recognize the rich legacy of Indigenous strengths, work alongside the patient to advance Indigenous health equity, and acknowledge the dominance of Western health models.” Please see lines 341-345. 

Acknowledging that each indigenous group has a unique culture and may have variations in structural issues contributing to complexity, how might an indigenous-centered PCAT generalize? Would it be applicable to all indigenous groups across the world, or would there need to be adaptations made to fit each group? This ties into the earlier comment about use of the already existing Collaborative Community Program in Remote Northern Territory.

It is indeed important to recognize the diversity within the global Indigenous populations, while acknowledging the common experiences of colonization that perpetuate health disparities. We have added a sentence under “additional considerations for complexity among Indigenous patients” which reads as follows: “Furthermore, PCATs for Indigenous peoples should abstain from making Pan-Indigenous assumptions but rather give space to respect diversity within Indigenous groups.” Please see lines 330-331.  

Minor: Please review for typos, missing or repeated words, etc.

Thank you, we have reviewed the entire manuscript for spelling/grammar.

---

## [Decision Letter · Decision Letter 1]

17 Aug 2022

Patient complexity assessment tools containing inquiry domains important for Indigenous patient care: A scoping review.

PONE-D-21-39743R1

Dear Dr. Sehgal,

We’re pleased to inform you that your manuscript has been judged scientifically suitable for publication and will be formally accepted for publication once it meets all outstanding technical requirements.

Kind regards,

Anand Nayyar, Ph.D.

Academic Editor

PLOS ONE

Additional Editor Comments (optional):

The Research Paper stands Accepted with no further revisions.

Reviewers' comments:

Reviewer's Responses to Questions

**Comments to the Author**

1. If the authors have adequately addressed your comments raised in a previous round of review and you feel that this manuscript is now acceptable for publication, you may indicate that here to bypass the “Comments to the Author” section, enter your conflict of interest statement in the “Confidential to Editor” section, and submit your "Accept" recommendation.

Reviewer #2: All comments have been addressed

2. Is the manuscript technically sound, and do the data support the conclusions?

Reviewer #2: Yes

3. Has the statistical analysis been performed appropriately and rigorously? 

Reviewer #2: Yes

4. Have the authors made all data underlying the findings in their manuscript fully available?

Reviewer #2: Yes

5. Is the manuscript presented in an intelligible fashion and written in standard English?

Reviewer #2: Yes

6. Review Comments to the Author

Reviewer #2: Thank you for the chance to re-review your manuscript. The authors have satisfactorily addressed my comments and I believe it is suitable for publication in PLOS ONE.

7. PLOS authors have the option to publish the peer review history of their article (what does this mean?). If published, this will include your full peer review and any attached files.

Reviewer #2: No

---

## [Editor Report · Acceptance letter]

22 Aug 2022

PONE-D-21-39743R1 

Patient complexity assessment tools containing inquiry domains important for Indigenous patient care: A scoping review. 

Dear Dr. Sehgal:

I'm pleased to inform you that your manuscript has been deemed suitable for publication in PLOS ONE. Congratulations! Your manuscript is now with our production department. 

Kind regards, 

on behalf of

Dr. Anand Nayyar 

Academic Editor

PLOS ONE